

**A dataset of microclimate and radiation and energy fluxes from the Lake Taihu Eddy**
**Flux Network**
Zheng Zhang[a,b], Mi Zhang[a,b,d], Chang Cao[a,b], Wei Wang[a,b], Wei Xiao[a,b,d], Chengyu Xie[a,b],
Haoran Chu[a,b], Jiao Wang[a,b], Jiayu Zhao[a,b], Lei Jia[a,b], Qiang Liu[a,b], Wenjing Huang[a,b],
Wenqing Zhang[a,b], Yang Lu[a,b], Yanhong Xie[a,b], Yi Wang[a,b], Yini Pu[a,b], Yongbo Hu[a,b], Zheng
Chen[a,b], Zhihao Qin[a,b], Xuhui Lee[c*]
a Yale-NUIST Center on Atmospheric Environment, International Joint Laboratory on
Climate and Environment Change (ILCEC), Nanjing University of Information Science and
Technology, Nanjing, Jiangsu Province, China
b Key Laboratory of Meteorological Disaster, Ministry of Education and Collaborative
Innovation Center on Forecast and Evaluation of Meteorological Disasters, Nanjing
University of Information Science and Technology, Nanjing, Jiangsu Province, China
c School of Forestry and Environmental Studies, Yale University, New Haven, CT, USA
d NUIST-Wuxi Research Institute, Wuxi, Jiangsu Province, China
* Corresponding author: xuhui.lee@yale.edu



**Abstract**
Eddy covariance data are widely used for the investigation of surface-air interactions.
Although numerical datasets exist in public depositories for upland ecosystems, few research
groups have released eddy covariance data collected over lakes. In this paper, we describe a
dataset from the Lake Taihu Eddy Flux Network, a network consisting of seven lake sites and
one land site. Lake Taihu is the third largest freshwater lake (area 2,400 km$^2$) in China, under
the influence of subtropical climate. The dataset spans the period from June 2010 to
December 2018. Data variables are recorded at half-hourly intervals and include
micrometeorology (air temperature, humidity, wind speed, wind direction, rainfall, and
water/soil temperature profile), the four components of surface radiation balance, friction
velocity, and sensible and latent heat fluxes. Except for rainfall and wind direction, all other
variables are gap-filled, with each datapoint marked by a quality flag. Several areas of
research can potentially benefit from the publication of this dataset, including evaluation of
mesoscale weather forecast models, development of lake-air flux parameterizations,
investigation of climatic controls on lake evaporation, validation of remote sensing surface
data products, and global synthesis on lake-air interactions. The dataset is publicly available
at https://yncenter.sites.yale.edu/data-access and from Harvard Dataverse
https://doi.org/10.7910/DVN/HEWCWM (Zhang et al., 2020)


## 1. Introduction

Inland lakes and reservoirs are a vital freshwater resource for the society. Globally, there are

more than 27 million water bodies with size greater than 0.01 km$^2$, occupying a total of 3.5%

of the Earth's land surface area (Downing et al., 2006; Verpoorter et al., 2014). Accurate

observation of the lake microclimate and lake-air interactions will help to better manage this

water resource and to better predict how it may be affected by environmental changes.

Towards that end, an increasing number of studies have deployed the eddy covariance (EC)

methodology to monitor physical state (temperature, wind, humidity) and process variables

(momentum flux, and radiation and energy fluxes) in the lake environment (Vesala et al.,

2006; Blanken et al., 2011; Nordbo et al., 2011; Wang et al., 2014; Li et al., 2015; Yusup and

Liu, 2016; Du et al., 2018; Hamdani et al., 2018; Xiao et al., 2018; Wang et al., 2019). Unlike

EC studies in upland ecosystems, however, data from these lake studies are rarely published

as data papers or are archived in public data depositories accessible by the broader scientific

community. For example, of the nearly 500 sites that have contributed EC and

micrometeorological data to AmeriFlux, a public data depository

(https://ameriflux.lbl.gov/data/data-availability/), none is a lake site. Although a few

scientific groups have provided data supplements to their scientific papers on lake-air fluxes

(e. g., Charusombat et al., 2018; Franz et al., 2018; Zhao and Liu, 2018), we are not aware of

a data paper devoted to systematic description and archival of EC lake observations.

In this paper, we describe the dataset from the Lake Taihu Eddy Flux Network (Lee et al.,

2014). Established in 2010, the network currently consists of six active lake sites, one



inactive lake site, and one active land site. Lake Taihu is the third largest freshwater lake
(area 2,400 km$^2$) in China. Data variables are recorded at half-hourly intervals and the
measurement has continued for over eight years. Several areas of research can potentially
benefit from the publication of this dataset, including evaluation of mesoscale weather
forecast models, development of lake-air flux parameterizations, investigation of climatic
controls on lake evaporation, validation of remote sensing surface data products, and global
synthesis on lake-air interactions.

This paper is organized as follows. Section 2 is a brief overview of the sites and the
instruments used by the network. This is followed, in Section 3, with a description of data
quality measures deployed during the field monitoring. Section 4 provides the essential
information about the dataset, including data variables, gap-filling methods, and data quality
flags. Results of post-field evaluation of the data quality are given in Section 5.

Users of this dataset may be interested in the relevant papers published by our group. Lee et
al. (2014) gave an overview of the Lake Taihu Eddy Flux Network. Using the data collected
at a subset of the sites and during the early phase of the network, Wang et al. (2014)
investigated the spatial variability of energy and momentum fluxes across the lake. Xiao et al.
(2013) improved the bulk parameterizations of heat, water and momentum fluxes for shallow
lakes. Deng et al. (2013) and Hu et al. (2017) modified the CLM lake simulator (Subin et al.,
2012) to improve its prediction of the lake evaporation. Wang et al. (2017) and Zhang et al.
(2019b) evaluated the performance of two mesoscale models of the lake-land breeze. More



recently, Xiao et al. (2020, manuscript in review) investigated drivers of the interannual
variability of the lake evaporation observed at one of the lake sites (BFG). The value of the
dataset is enhanced by these peer-reviewed publications because they have helped us to
continuously improve our measurement and data processing protocols. For example, we have
used the locally-calibrated bulk parameterizations of Xiao et al. (2013) to gap-fill the flux
variables.

**2. Sites and Instrumentation**
**2.1 Sites and data periods**
Table 1 shows the basic site information and Figure 1 is a map that gives the relative position
of Lake Taihu in China and locations of the EC measurement sites. Also shown in Figure 1
are WMO baseline weather stations around the lake, whose data can be obtained from
National Meteorological Information Center in China (http://data.cma.cn/site/index.html).
The lake, located between the latitudinal range of 30˚5′40″ N to 31˚32′58″ N and
longitudinal range of 119˚52′32″ E to 120˚36′10″ E, has a total area of 2400 km$^2$ and an
average depth of 1.9 m. The climate is subtropical monsoon, with an annual mean
temperature of 16.2°C and annual total precipitation of 1122 mm. The lake is ice-free
throughout the year.

The EC network consists of seven lake sites and one land site. The lake sites (Meiliangwan
(MLW), Dapukou (DPK), Bifenggang (BFG), Xiaoleishan (XLS), Pingtaishan (PTS),
Dongtaihu (DTH), Meiliangwan2 (MLW2)) are distributed according to biological





characteristics and across eutrophication gradients of the lake. The MLW site, located in
Meiliangwan Bay near the north shore of Lake Taihu, was the first site in operation; the
measurement began in June 2010 and was replaced by MLW2 in 2018, at 10 km southwest of
MLW. Both MLW and MLW2 sites are located in the lake eutrophic zone. BFG is located in
the east part of Lake Taihu in relatively clean water inhabited by submerged vegetation with
a growth season from April to November. DTH is located in the shallow water (mean depth
of 1.3 m) in the southeast part of the lake. After more than 20 years of crab aquaculture, this
zone was returned to unmanaged state in December 2018 in order to improve water quality.
The observation at DTH enables the examination of lake-air exchange processes in the
transition from human management to a natural state. PTS is situated in the middle of Lake
Taihu where occasional algal blooms occur and no aquatic vegetation is present. DPK is
located near the west shore, in a relatively deep (depth 2.5 m) super eutrophic zone due to
heavy influence of agricultural and urban runoffs. XLS is located in the relatively clean and
vegetation-free zone in the southeast. Finally, DS is a land site surrounded by rice agriculture,
serving as a land reference for the lake sites. The MLW site is situated at a distance of 200 m
from the north shore of the lake. All the other lake sites in the lake are at a distance of more
than 1 km away from the land.

**2.2 Instrumentation**
Each site is equipped with an EC system for long-term, continuous monitoring of the surface
momentum, sensible heat, latent heat and carbon dioxide fluxes. The EC system consists of a
sonic anemometer/ thermometer (Model CSAT3A; Campbell Scientific, Logan, UT, USA)





and a $CO_2/H_2O$ infrared analyzer (Model 7500A, LI-COR, Inc., Lincoln, NE, USA at DS,
MLW, MLW2 and DPK; Model EC150, Campbell Scientific at other sites). The EC
instrument is at a height of 3.5 to 20 m above the water or the soil surface. Other
measurements include air humidity and air temperature (Model HMP45D/HMP155A;
Vaisala, Inc, Helsinki, Finland), wind speed and wind direction (Model 03002; R. M. Young
Company, Traverse City, MI, USA) and four components of the net radiation (Model CNR4;
Kipp & Zonen B. V., Delft, the Netherlands). At the lake sites, water temperature profile was
measured with temperature probes (Model 109-L; Campbell Scientific) at the water depth of
20, 50, 100, and 150 cm and in the sediment at about 5 cm below bottom of the water column.
At the DS land site, soil temperature profile was measured with the same type of probes at
the depths of 5, 10, 20 cm. The MLW and the DS sites are supported by A/C power and other
sites are powered by battery packs connected to solar panels.

The methane flux was measured at MLW, BFG and DTH for selected periods, in addition to
the standard variables described above, using a flux-gradient system (at MLW; Xiao et al.
2014) and an open-path eddy covariance system (at BFG and DTH, Zhang et al., 2019a). The
carbon dioxide and methane flux data are not included in the current version of the data
release but will be added at a later time after the data quality has been fully examined and the
data gaps filled.

All the variables are reported as 30-min averages. The EC covariance data are expressed in
the natural coordinate system (Lee et al., 2004). Additionally, a small density correction has



been applied to the water vapor flux according to Webb et al. (1980).

**3 Data Quality Control during Field Monitoring**
Every site in the Lake Taihu Eddy Flux Network is equipped with a wireless transmission
module for real-time monitoring and for data transmission. Time series of all 30-min
variables are examined weekly and abnormal behaviors are flagged for site operators. Each
site is visited every one to two months to perform instrument repair and maintenance and to
download 10 Hz EC data. The data coverage rates are summarized in Table 2, where the
percentage values represent the proportions of data with quality flag 0 (Table 3).

The four-way net radiometers at MLW and XLS were compared in the field against a
laboratory standard of the same type in the summer of 2018 to check their long-term stability.
These two sites were chosen because they have been in operation for more than five years.
Additionally, the radiometer at MLW was relocated to MLW2 after MLW had been
discontinued. The laboratory standard, which had been calibrated at the manufacturer prior to
this performance evaluation, was mounted next to the field instrument for about 10 days at
each site, covering overcast to clear-sky conditions. The mean bias error was smaller than 1
W m$^{-2}$ for all the radiation components. It was -0.81, -0.81, 0.79 and -0.44 W m$^{-2}$ for the
downward shortwave, upward shortwave, downward longwave and upward longwave
radiation flux at MLW, respectively. The corresponding values were 0.91, 0.40, 0.69 and
0.77 W m$^{-2}$ for XLS. (Comparison experiments are being planned for the other sites.)



The EC gas analyzers were calibrated every one to two years. The zero-point calibration was
carried out with high-purity nitrogen gas, the $CO_2$ span calibration was made with standard
carbon dioxide gases (in the concentration range of 389 to 525 ppm) provided by the National
Institute of Meteorology (NIM), China and certified to an accuracy of 1%, and the $H_2O$ span
calibration was made with a portable dew-point generator (LI-610; LI-COR, Inc.).

**4. Gap-filling Methods and Data Quality Flags**
We use five-point moving average to screen outliers. If the deviation from the moving
average is greater than two standard deviations, the data point is discarded. If a gap length is
30 min to 1 h, the gap is filled by linear interpolation. Larger gaps in meteorological variables,
radiation components and water temperature are filled with linear regression involving
observation of the same variable at another site. This spatial interpolation consists of three
steps. First, linear correlation is calculated using the valid data at the target site and at all
other sites for the month during which the data gap occurred. Second, the observation at the
site with the highest linear correlation is used to establish a linear regression equation. Third,
the gap at the target site is filled with the linear regression and the observation at that site.

Radiation data gaps at the DS land site require special treatment. The radiometer at DS eddy
flux site ended in January 2013. Subsequent measurements of the radiation component are
provided by a radiometer belonging to the Dongshan WMO weather station at a distance of
50 m from the eddy covariance tower (Figure 1). While large gaps in meteorological
variables (air temperature, relative humidity, wind speed and air pressure), downward solar



radiation and downward longwave radiation are filled with the spatial interpolation method,
large gaps in upward shortwave radiation and upward longwave radiation cannot be filled
with data from other lake sites even with linear regression. In the case of the upward
shortwave radiation, the data gaps were filled using the relationship between downward
shortwave radiation and the monthly mean albedo. In the case of upward longwave radiation,
the data gaps were filled by a regression equation between the upward longwave radiation
and the fourth power of soil temperature at 5-cm depth. Compared to the original data, the
gap-filled data do not capture the full diurnal variations but the daily-mean upward shortwave
and longwave radiation fluxes seem reasonable.

Large data gaps in the EC variables (sensible heat flux, latent heat flux and friction velocity)
are filled with a hybrid method. If observations exist for the relevant state variable, the gap is
filled with the bulk transfer relationship using a locally-tuned transfer coefficient (Xiao et al.,
2013). For example, the relationship for filling gaps in the sensible heat flux is
$$H = \rho_a c_p C_H U (T_s - T_a)$$
where $\rho_a$ is air density, $c_p$ is specific heat of air at constant pressure, $C_H$ is the transfer
coefficient for sensible heat, $T_a$ is air temperature and $T_s$ is water surface temperature. If data
for the state variable is missing, the spatial interpolation method is used to fill the gaps in
these EC variables.

The spatial interpolation method described above occasionally causes a sudden jump at the
beginning or end of a data gap. To harmonize the data, we apply a 5-point moving averaging



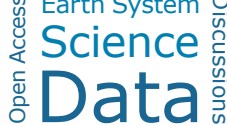

to the gap-filled time series. If a data point deviates by 2 times of the standard deviation from
the moving average, it is replaced by linear interpolation using the two adjacent data points.

Each data point is assigned a quality flag to distinguish original measurements and gap-filled
values and gap-filling methods (Table 3). Flag 0 indicates high-quality original data. Other
flag values indicate gap-filled data or missing values. Flag 1 indicates that the data was filled
by temporal interpolation. Flag 2 indicates that the data was filled by the spatial interpolation
method. Flag 3 for the EC variables indicates that the data was filled by the bulk relationship.
We also use Flag 3 to mark the upward shortwave and longwave radiation data filled with the
albedo and the surface temperature relationship, respectively, for the DS land site. Missing
values occur on some situations, which are marked with Flag 4. Figure 3 is an example
showing the gap-filled time series of several variables at BFG along with the flag status.

Rainfall data has not been quantity-controlled or gap-filled. Because of the episodic nature of
rainstorms and high spatial variability of rainfall, it is not appropriate to fill data gaps with
the time or spatial interpolation method. The total rain amount is likely biased low because
no wind screens are used to protect the rain gages from the influence of wind which is much
higher on the lake than on land (Figure 4 below). On several site visits, the drain opening to
the tipping bucket was found to be partially blocked by debris. Rain amount at a constant and
low rate and excessively long rain duration are evidence of such blockage. The flag status of
0 for the rainfall variable simply indicates that the field measurement is available, but it does
not guarantee high data quality.

The data coverage begins from the start time of each site (Table 4) and the ends in December
2018. The time resolution is 30 min. The dataset includes microclimate variables (air pressure,
air temperature, relative humidity, wind speed, wind direction and rainfall), radiation fluxes
(upward and downward shortwave radiation, upward and downward longwave radiation),
water temperature at depth of 0.2 m, 0.5 m, 1.0 m and1.5 m, and in the 5-cm sediment) and
eddy fluxes (friction velocity, and sensible heat and latent heat fluxes; Table 4). The time
stamp is Beijing time (UTC + 8 h) given by data columns 1 to 5 as year, month, day, hour,
and minute, and marks the end of the observation period. For example, time stamp "2012, 1,
12, 00" indicates that the data acquisition period is from 11:30 to 12:00 on January 1, 2012.

Although the data table does not include the radiative surface temperature $T_s$, the user can
easily calculate it from the two longwave radiation fluxes, as
$$T_s = (\frac{L_\uparrow - (1-\varepsilon)L_\downarrow}{\varepsilon\sigma})^{\frac{1}{4}}$$

where σ is the Stefan-Boltzmann constant, $\varepsilon$ is emissivity, and $L_\uparrow$ and $L_\downarrow$ are upward and
downward longwave radiation flux, respectively. We use a value of 0.97 for lake surface
emissivity in this calculation (Deng et al., 2013; Wang et al., 2014).

**5. Data Consistency Evaluation**
Figure 4 compares the annual mean air temperature, relative humidity, and wind speed at the
Taihu eddy flux sites with those at the four WMO weather stations (Wuxi, Liyang, Huzhou





and Dongshan) around the lake (Figure 1). The error bars represent the maximum and
minimum values among the four WMO stations and the lines represent the mean values of
the four station measurements. The annual mean air temperature at DTH is 0.3°C higher than
the station mean. At other sites, air temperature is in close agreement of the weather station
data, in terms of both magnitude and inter-annual variability. The annual mean wind speed at
MLW, a site near the shoreline, is comparable with the station data. At other more exposed
sites, the wind speed is much higher than observed at the WMO stations. The annual mean
relative humidity RH shows a larger spread among the eddy flux sites than among the WMO
stations partly because the measurement height at the eddy flux sites is not standardized
(Table 1). The upward trends in RH over time at DPK and XLS seem to be related more to
aging of the sensor than to a real inter-annual variability. We have not fully investigated this
aging problem, but it is possible to rectify it by doing a detailed regression analysis against
the station data.

Consistency of the energy flux variables can be evaluated with the energy balance closure.
Using observations made at a subset of the sites in the earlier years of the flux network,
Wang et al. (2014) reported a closure rate of 70 % to 110 % on the monthly basis, meaning
that the sum of the measured monthly sensible and the latent heat flux $H + \lambda E$ is 70 % to
110 % of the monthly available energy $R_n - G$, where $R_n$ net radiation and $G$ is heat storage
in the water column. By selecting days without data gaps, we found that the daily energy
balance closure is in the range between 66 % and 78 % for all the lake sites and all the years.
Such closure rates are typical of eddy covariance observations (Tanny et al., 2008; Wilson et



al., 2002).

We have shown that the monthly latent heat flux at the lake sites MLW, BFG and DPK
during July 2010 to August 2012 follows the Priestley-Taylor (PT) model prediction with the
original PT constant α of 1.26 and that at the DS land site is in agreement with the PT model
if the constant is lowered to 1.0 (Lee et al., 2014). Figure 5 demonstrates that the same
relationships hold for all the sites and all the observational months, indicating the overall
stability of our measurement systems and the robustness of our gap-filling procedure. The
reader is reminded that the monthly latent heat flux in Figure 5 has been adjusted to force
energy closure following the method recommended by Barr et al. (1994), Blanken et al.
(1997) and Twine et al. (2000). (The half-hourly flux data in the data archive have not been
adjusted for energy balance.)

The Stefan-Boltzmann Law offers another way for checking data consistency. Because the
lake surface emits longwave radiation like a blackbody and because the annual mean air
temperature and the surface water temperature are nearly identical at this lake (Wang et al.,
2014), the change in the annual upward longwave radiation $\Delta L_\uparrow$ can be expressed as
$$\Delta L_\uparrow = 4\sigma T_a^{\,3} \Delta T_a$$
where $T_a$ is annual mean air temperature, and $\Delta$ is the difference between the target year and
the year with the lowest air temperature observed at the site. All the five long-term lake sites
show good consistency between the longwave radiation and the air temperature observations
(Figure 6).




## 6 Data availability

All data can be open-accessed online for download and use at https://yncenter.sites.yale.edu/
and from Harvard Dataverse (https://doi.org/10.7910/DVN/HEWCWM, Zhang et al., 2020).

## 7 Summary

The dataset described here consists of microclimate variables (air temperature, air humidity,
wind speed, wind direction, water or soil temperature profile, and rainfall), four components
of the radiation balance, friction velocity, and sensible and latent heat fluxes observed at
seven lake sites and one land site. The period of coverage is from June 2018 to December
2018. The observation interval is 30 min. Except for rainfall and wind direction, all other
variables have been gap-filled. Every data point is tagged with a data quality flag to help the
user determine how to best use the data.

## Author contribution

XL, WX and MZ directed the field program, ZZ performed data gap-filling and prepared the
data for public release, CC, WW, CX, HC, JW, JZ, LJ, QL, WH, WZ, YL, YX, YW, YP, YH,
ZC and ZQ participated in field data collection, and ZZ, XL and MZ wrote the manuscript.

## Competing interests

The authors declare no conflict of interest.

## Acknowledgments



This work was supported by the National Key R&D Program of China (2019YFA0607202),
the National Natural Science Foundation of China (grant numbers 41575147, 41801093, and
41475141) and  the Priority Academic Program Development of Jiangsu Higher Education
Institutions (grand number PAPD).






**Table 1.** A list of sites in the Lake Taihu Eddy Flux Network

| Site ID | MLW | DPK | BFG | XLS | PTS | MLW2 | DTH | DS |
|---|---|---|---|---|---|---|---|---|
| Lat (°N) | 31.4197 | 31.2661 | 31.1685 | 30.9972 | 31.2323 | 31.3818 | 31.0611 | 31.0799 |
| Long (°E) | 120.2139 | 119.9312 | 120.3972 | 120.1344 | 120.1086 | 120.1608 | 120.4704 | 120.4346 |
| Start date | Jun 2010 | Aug 2011 | Dec 2011 | Nov 2012 | Jun 2013 | Feb 2018 | Nov 2017 | Apr 2011 |
| Biology | Eutrophic | Super eutrophic | Submerged macrophyte | Transitional | Mesotrophic | Eutrophic | Aquaculture | Cropland/ Rural residence |
| Met height (m) | 3.5 | 8.0 | 8.5 | 9.4 | 8.5 | 6.0 | 4.5 | 10.0 |
| $T_w$ / $T_s$ depths (cm) | 20, 50, 100, 150, sediment | 20, 50, 100, 150, sediment | 20, 50, 100, 150, sediment | 20, 50, 100, 150, sediment | 20, 50, 100, 150, sediment | 20, 50, 100, 150, sediment | 20, 50, sediment | 5, 10, 20 |
| Radiation height (m) | 1.5 | 1.5 | 1.5 | 1.5 | 1.5 | 1.5 | 1.5 | 3.0 |
| EC height (m) | 3.5 | 8.5 | 8.5 | 9.4 | 8.5 | 6.0 | 4.5 | 20 |







**Table 2.** Percent of data coverage

| Variable type | MLW | DPK | BFG | XLS | PTS | DTH | MLW2 | DS |
|---|---|---|---|---|---|---|---|---|
| Micrometeorology | 93.3 | 81.1 | 97.6 | 97.0 | 97.5 | 98.1 | 90.3 | 91.7 |
| Radiation flux | 85.5 | 90.8 | 96.9 | 97.4 | 98.6 | 98.2 | 98.2 | 82.7 |
| Water/soil temperature | 83.4 | 81.3 | 94.0 | 91.1 | 90.3 | 87.7 | 22.4 | 98.4 |
| Eddy flux | 73.3 | 61.8 | 82.7 | 79.1 | 80.6 | 85.7 | 85.5 | 82.8 |










**Table 3.** A list of data quality flags

| Flag | Data quality description |
|------|--------------------------|
| 0 | Original data |
| 1 | Gap-filled with time interpolation |
| 2 | Gap-filled with spatial interpolation |
| 3 | Gap-filled with bulk relationship |
| 4 | NAN |








**Table 4.** A list of data columns and variable definitions

| Column | Description | Variable name | Unit |
|:---:|:---:|:---:|:---:|
| 1 | Year | Year | − |
| 2 | Month | Month | − |
| 3 | Day | Day | − |
| 4 | Hour | HH | − |
| 5 | Minute | MM | − |
| 6 | Day of Year | DOY | − |
| 7 | Air pressure | P | kPa |
| 8 | Quality flag of air pressure | P_flag | |
| 9 | Air temperature | Ta | °C |
| 10 | Quality flag of air temperature | Ta_flag | |
| 11 | Relative humidity | RH | % |
| 12 | Quality flag of Relative humidity | RH_flag | |
| 13 | Wind speed | WS | $\mathrm{m\ s^{-1}}$ |
| 14 | Quality flag of wind speed | WS_flag | |
| 15 | Wind direction | WD | Degree |
| 16 | Quality flag of wind direction | WD_flag | |
| 17 | Rainfall | R | mm |
| 18 | Quality flag of rainfall | R_flag | |
| 19 | Upward shortwave radiation | UR | $\mathrm{W\ m^{-2}}$ |
| 20 | Quality flag of upward shortwave radiation | UR_flag | |
| 21 | Downward shortwave radiation | DR | $\mathrm{W\ m^{-2}}$ |
| 22 | Quality flag of downward shortwave radiation | DR_flag | |
| 23 | Upward longwave radiation | ULR | $\mathrm{W\ m^{-2}}$ |
| 24 | Quality flag of upward longwave radiation | ULR_flag | |
| 25 | Downward longwave radiation | DLR | $\mathrm{W\ m^{-2}}$ |
| 26 | Quality flag of downward longwave radiation | DLR_flag | |
| 27 | Water temperature at 0.2 m | $T_w\_20$ | °C |




| 28 | Quality flag of Water temperature at 0.2 m | $T_w\_20\_flag$ | |
| 29 | Water temperature at 0.5 m | $T_w\_50$ | °C |
| 30 | Quality flag of Water temperature at 0.5 m | $T_w\_50\_flag$ | |
| 31 | Water temperature at 1.0 m | $T_w\_100$ | °C |
| 32 | Quality flag of Water temperature at 1.0 m | $T_w\_100\_flag$ | |
| 33 | Water temperature at 1.5 m | $T_w\_150$ | °C |
| 34 | Quality flag of water temperature at 1.5 m | $T_w\_150\_flag$ | |
| 35 | Sediment temperature | $T_w\_bot$ | °C |
| 36 | Quality flag of sediment temperature | $T_w\_bot\_flag$ | |
| 37 | Friction velocity | $U^*$ | m s$^{-1}$ |
| 38 | Quality flag of friction velocity | $U^*\_flag$ | |
| 39 | Sensible heat flux | H | W m$^{-2}$ |
| 40 | Quality flag of sensible heat flux | H_flag | |
| 41 | Latent heat flux | LE | W m$^{-2}$ |
| 42 | Quality flag of latent heat flux | LE_flag | |

Notes: 1) Time marks end of an observation in Beijing time (UTC+8:00); 2) At the DS site, columns 27, 29,
and 31 represent soil temperature at 5, 10, and 20 cm, respectively.
352  .





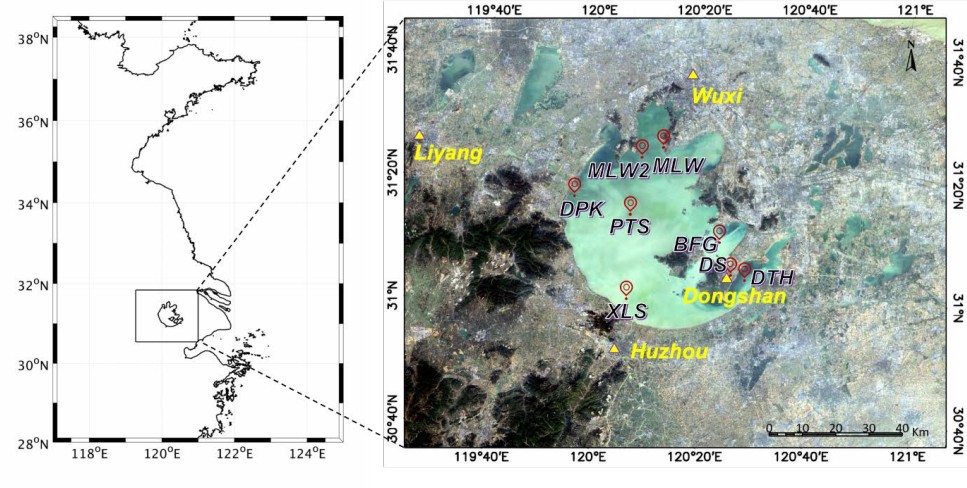


**Figure 1.** Map showing locations of Lake Taihu, eddy covariance sites (red bubbles) and

WMO weather stations (yellow triangles). The background is a natural color image from

LANDSAT 8 without correction for atmospheric interference.








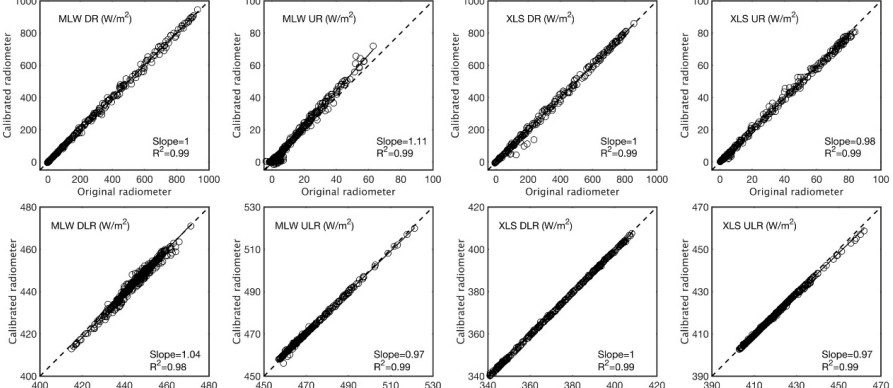


**Figure 2.** Comparison of four components of the radiation balance between the original

radiometer (horizontal axis) and a laboratory standard (vertical axis) at MLW and XLS. Refer

to Table 4 for variable definitions.





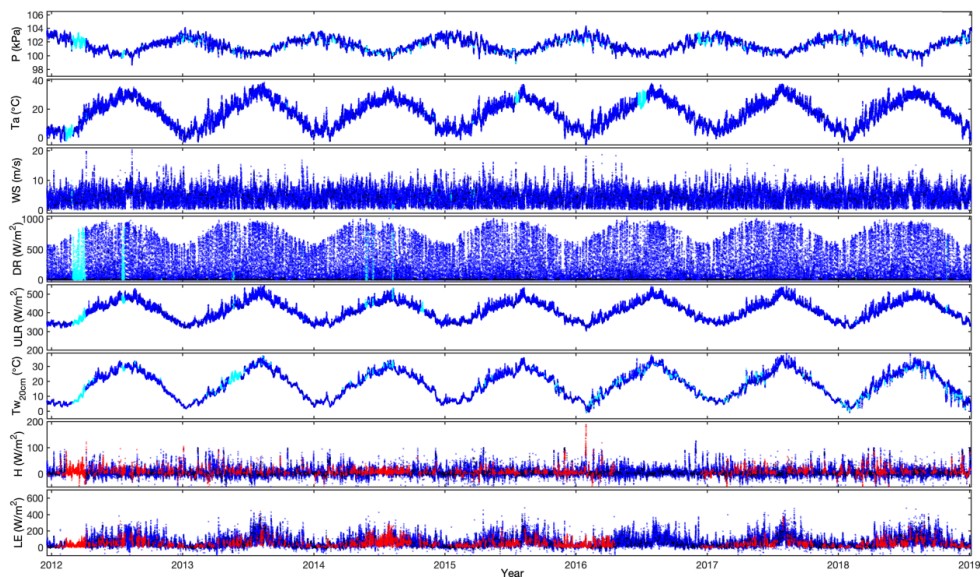


**Figure 3.** Complete gap-filled time series for selected variables observed at BFG. Blue, black, cyan and red dots represent quality flag 0, 1, 2, and 3, respectively. Variable definitions are given in Table 4






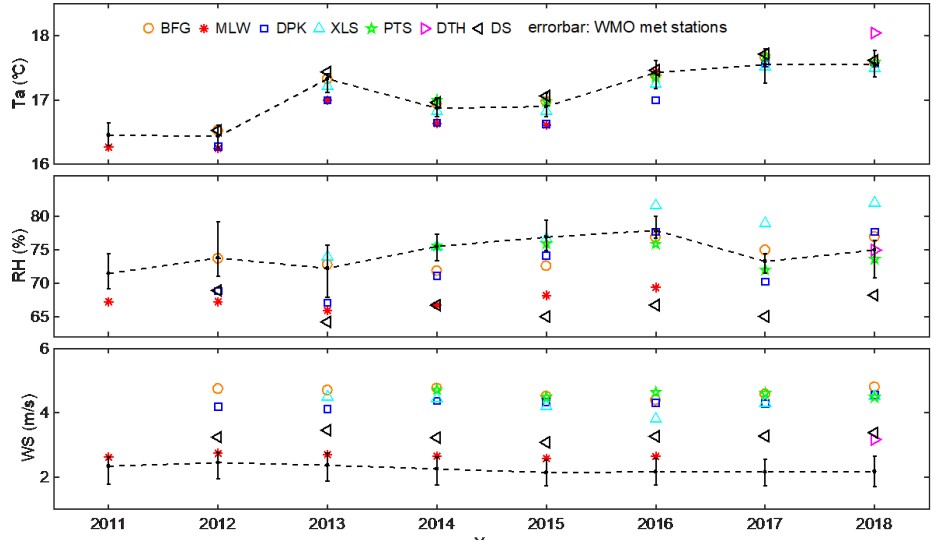


**Figure 4.** Annual mean air temperature (top), relative humidity (middle) and wind speed
(bottom) observed at the eddy flux sites (symbols) and at the four WMO weather stations
around the lake (line). Error bar represents the range of the annual means of the four WMO
stations.








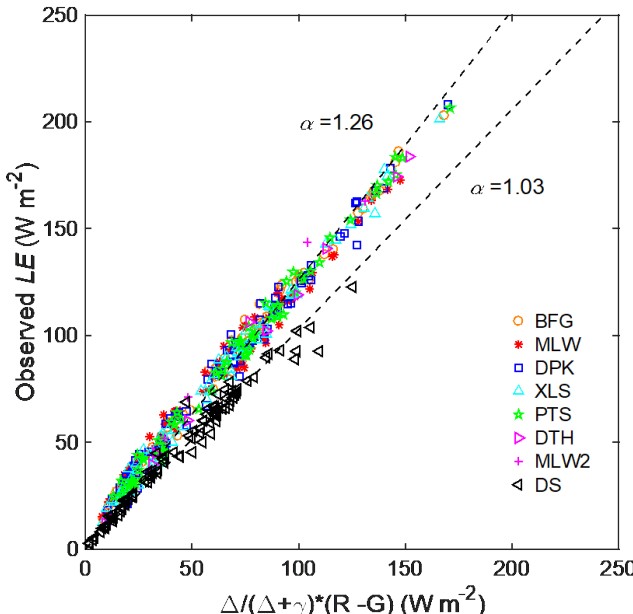


**Figure 5.** Comparison of observed monthly latent heat flux with Priestley-Taylor model

prediction using the origional α coefficeint of 1.26 and a modified coefficient of 1.03. Here

$R_n$ is net radiation, $G$ is heat storage in the water column, $\Delta$ is the slope of the saturation

vapor pressure curve, and $\gamma$ is the psychrometric constant.




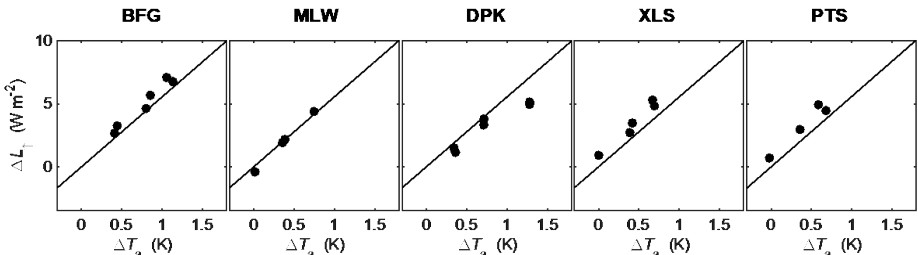


**Figure 6.** The relatinship between changes in observed annual mean upward longwave

radiation flux and annual mean air temperature (dots). Solid lines represent the prediction of

the Stefan-Boltzmann Law.




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
