# Peer review of "A dataset of microclimate and radiation and energy fluxes from the Lake Taihu Eddy Flux Network"

_Earth System Science Data, 2020_

## Referee Comment (RC1) · Anonymous Referee #1 · 5 Jul 2020

The manuscript describes a dataset measured by the Lake Taihu Eddy Flux Network which consists of seven eddy covariance (EC) flux stations over Lake Taihu and one EC flux tower over the land as a reference. Although EC flux measurements over inland waters have increased worldwide, such a flux network over a single lake is rare. Besides the uniqueness of the dataset stated in the manuscript, it could provide a valuable perspective in terms of spatial variability in fluxes and associated controlling factors. The dataset should benefit to broader communities in micrometeorology, hydrology, remote sensing, water resource managements, and modeling to name a few. The descriptions of the sites, instruments, and methods are clear and adequate. The dataset is of high quality as also reflected by their published research articles (I read most of them in the past). The manuscript is well written and structured. I would

recommend its publication.

Here are a few minor comments:

I understand water levels vary, but would it be good to provide the lake bathymetry also with the tower locations to give users a better understanding of the lake?

Do you have water level measurements or provide information that helps users to find how water levels vary?

Lines 130-131: This may be a little bit misleading since the heights for the lake sites vary from 3.5 to 9.4 m and the EC height for the land site is 20 m.

Line 137: please explain how you keep these sensors in such fixed depths.

---

## Referee Comment (RC2) · Anonymous Referee #2 · 7 Jul 2020

[referee-annotated manuscript omitted]

---

## Author Comment (AC1) · 28 Jul 2020

**Response to Review 1**
(Review comments in Italic, response in upright Roman)

*The manuscript describes a dataset measured by the Lake Taihu Eddy Flux Network which consists of seven eddy covariance (EC) flux stations over Lake Taihu and one EC flux tower over the land as a reference. Although EC flux measurements over inland waters have increased worldwide, such a flux network over a single lake is rare. Besides the uniqueness of the dataset stated in the manuscript, it could provide a valuable perspective in terms of spatial variability in fluxes and associated controlling factors. The dataset should benefit to broader communities in micrometeorology, hydrology, remote sensing, water resource managements, and modeling to name a few. The descriptions of the sites, instruments, and methods are clear and adequate. The dataset is of high quality as also reflected by their published research articles (I read most of them in the past). The manuscript is well written and structured. I would recommend its publication.*

Thank you.

*Here are a few minor comments:*
*I understand water levels vary, but would it be good to provide the lake bathymetry also with the tower locations to give users a better understanding of the lake?*

*Do you have water level measurements or provide information that helps users to find how water levels vary?*

The daily water level is monitored by the TAIHU BASIN AUTHORITY at five locations around the lake (http://www.tba.gov.cn/slbthlyglj/sj/sj.html), as shown in Figure R1 below. Using this time series, we have constructed the water depth for our eddy covariance sites (Figure R2, added to the revision as Figure 2).

[Figure]

**Figure R1.** Daily water level (above local sea level) in Lake Taihu

[Figure]

**Figure R2** Water depth at the eddy covariance sites

We have added the following text to the revision:
"The lake water level is monitored daily by the Taihu Basin Authority (http://www.tba.gov.cn/) at five locations around the lake. Using the water-level time series, we have constructed the water depth for our eddy covariance sites (Figure 2)."

*Line 130-131: This may be a little bit misleading since the heights for the lake sites vary from 3.5 to 9.4 m and the EC height for the land site is 20 m.*

Thank you. This sentence has been changed to "The EC instrument is at a height of 3.5 to 9.4 m above the water surface at the lake sites and at a height of 20 m above the ground at the land site."

*Please explain how you keep these sensors in such fixed depths.*

We have added the following explanation:
"The top four sensors were tied to a nylon rope hanging from a buoy to ensure that they were at the designed depths regardless of water level fluctuations."

---

## Author Comment (AC2) · 28 Jul 2020

**Response to Review 2**
(Review comments in Italic, response in upright Roman)

*In this manuscript, the authors describe a significant database of eddy-covariance and micro-meteorological measurements over a large lake in China. The data is collected by an eddy flux network consisting of seven lake and one land eddy-covariance towers. Data is collected from 2010 to 2018. The data presented will be valuable for researchers in various disciplines like air-water interactions, climate and weather modeling, and water management. The manuscript is generally well written, and the authors presented the necessary information. I have several comments that the authors should consider upon revision of the manuscript. The comments are provided on the attached PDF file.*

Thank you for your careful review of our manuscript.

*Line 23: replace "upland" with "land"*
Done.

*Line 28: half-hourly intervals: I guess these are half-hourly averages.*
We have changed the sentence to "Data variables are saved as half-hourly averages…"

*Line 47 deployed: replace "deployed" with "employed"*
Done

*Line 52 upland: replace "upland" with "land"*
Done

*Line 73 deployed: replace "deployed" with "employed"*
Done

*Line 126 EC system for long-term monitoring: For the lake stations, you should explain briefly how did you install the stations. Did you install the sensors on stationary or floating platforms? Floating platform may be susceptible to oscillations due to water waves, which may affect sensors' readings.*

In response, we have added the following text for clarity
"Measurements at the lake sites were made on fixed platforms. Readers are referred to Lee et al. (2014) and Xiao et al. (2017) for photographs of the platform and the instruments."

*Line 129 infrared analyzer: add "gas" between "infrared" and "analyzer"*
Corrected

Line 131: instrument: replace "instrument" with "system"
Corrected

*Line 138 the DS land site: Did you measure soil heat flux at the land site?*

We did monitor soil heat flux at the 5-cm depth at DS. The data and data flag are added as columns 33 and 34. The dataset on Harvard Dataverse and on our website have been updated. Thank you.

*Line 142-147 If methane flux is not included in this paper, there is no need to provide all these details. I suggest removing this paragraph.*
Removed.

*Line 149 EC covariance: EC is eddy covariance so the word "covariance" appears twice.*
Corrected

**Q12:** *Line 150: natural coordinate system: Please provide a more specific explanation of this system, and its consequence for potential users of the data.*

We have added the following explanation:
"In this coordinate system, the longitudinal coordinate axis is aligned with the 30-min mean velocity vector so that the 30-min mean lateral and vertical velocity components are zero and the magnitude of the mean velocity is equal to the mean longitudinal component, and the covariance between the lateral and the vertical velocity components is zero."

*Line 159: proportions of data with quality flag 0: What is the meaning of this quality flag? Please give some explanation.*

We have expanded the sentence to
"… where the percentage values represent the proportions of data with quality flag 0, which indicates high-quality original measurement (Table 3)."

**Q14:** *Line 179: Data Quality Flags: The description of data quality flags is confusing since you do not refer to the common quality flags used in the eddy covariance methodology. The commonly used flags are related to the validity of steady state and turbulence development conditions. Please revise the text to better clarify this issue.*

For clarification, we have modified the text to
"Each data variable is assigned a quality flag to distinguish original measurements and gap-filled values and gap-filling methods (Table 3). The data flags employed here should not be confused with quality flags commonly assigned to the EC methodology in the literature. Specifically, Flag 0 indicates high-quality original data. Other flag values indicate gap-filled data or missing values."

**Q15:** *Line 201-202 "…Compared to the original data, the gap-filled data do not capture the full diurnal variations." Explain why?*

An explanation is added as
"Compared to the original data, the gap-filled data do not capture the full diurnal variations because the 5-cm soil temperature has smaller diurnal amplitudes than the soil surface temperature, but the daily-mean upward longwave radiation flux seems reasonable."

**Q16:** *Line 207 the bulk transfer relationship: I guess this relationship is extracted from another site where data is not missing? Please explain.*

This relationship is established for each site using data collected during periods when data on both the flux and the state variables were available. We have modified the text to improve clarity:

"The transfer coefficient $C_H$ is determined from the observed H and the state variables (U, $T_a$ and $T_s$) outside the gap periods. The missing data on H is then filled with the above relationship using the tuned $C_H$ the observed U, $T_a$ and $T_s$."

*Line 240: remove "the"*
Corrected.

*Line 240 (Table 4): replace "Table 4" with "Table 1"*
Corrected.

*Line 245: remove "and"*
Corrected.

*Line 247-248 "2012, 1, 12, 00": I think "month" or "day" are missing.*
We have changed it to "2012, 1, 1, 12, 00"

*Line 265: Which station?*
We have changed to: "the four WMO stations."

*Line 278: add "is" before "net"*
Corrected.

*Line 313: replace "2010" with "2018"*
Corrected.

**Q24:** *Line 339: Percent of data coverage: Explain more specifically what are these percentages.*

The table caption has been modified as: "**Table 2.** Percent of data coverage. The percentage represents the proportion of high-quality original measurement."

*Line 350: add "half-hourly" between "an" and "observation"*
Corrected.

*Line 359: In the map, I recommend adding names of some major cities of China for easier orientation of the lake location in China.*
Done.

*Line 359: Indicate which sites are on the lake and which is on the land.*
Done.

*368 Figure 2: This figure is not mentioned in the text.*
This figure is now cited in the main text.

*Line 387: replace "bar" with "bars"*
Corrected.

*Line 387: replace "represents" with "represent"*
Corrected.

*Line 387 (line): If the values are annual means, the lines connecting the points have no meaning. Please revise.*
We include lines to help better visualize inter-annual variability.

*Line 393 latent heat flux: This is not the observed LE as indicated on the Y-axis label, but LE adjusted for energy balance closure. This should be indicated.*

This is a very good point. We have changed the Y-axis label to "Monthly LE"

---

## Author Response (AR2)

Response to editor's comments (ms essd-2020-64)

*Editor's comments:*

*"Reader needs to see a much better, more explicit uncertainty analysis. What authors call 'data consistency evaluation' (Section 5) provides good validation, but users should also find detailed informed discussion of uncertainties: instrument uncertainties, performance / deployment uncertainties, environmental (weather, water height) uncertainties, etc. Authors mention many sources of uncertainty (sensor performance, for example) throughout the manuscript but never pull those uncertainties together into a composite uncertainty budget. Figure 3 shows high R2 values for radiation intercomparison, good. Figure 5 shows uncertainty bars but only for the WMO reference stations. Pull this information together into a table of estimated uncertainty by parameter measured, with a short section of author explanations and comments? Data as presented seem to lack any uncertainty, which authors and users will recognize as not plausible."*

Response:

Thank you for this suggestion. In response, we have added a Table (Table 5) to summarize the three types of uncertainty. We have added the following passages as explanation:

In section 2.2: "To evaluate the performance of field EC systems, we installed a closed-path EC system (Model CPEC200; Campbell Scientific) at BFG for a brief period in the summer of 2020. The performance uncertainty was based on the difference between the field open-path EC system and this closed-path system. "

[revised manuscript text omitted]